# Spin Freezing and Its Impact on Pore Size, Tortuosity and Solid State

**DOI:** 10.3390/pharmaceutics13122126

**Published:** 2021-12-09

**Authors:** Joris Lammens, Niloofar Moazami Goudarzi, Laurens Leys, Gust Nuytten, Pieter-Jan Van Bockstal, Chris Vervaet, Matthieu N. Boone, Thomas De Beer

**Affiliations:** 1Laboratory of Pharmaceutical Technology, Department of Pharmaceutics, Ghent University, Ottergemsesteenweg 460, B-9000 Gent, Belgium; joris.lammens@ugent.be (J.L.); chris.vervaet@ugent.be (C.V.); 2Department of Physics and Astronomy, Radiation Physics, Ghent University, Proeftuinstraat 86/N12, B-9000 Gent, Belgium; niloofar.moazamigoudarzi@ugent.be (N.M.G.); matthieu.boone@ugent.be (M.N.B.); 3Centre for X-ray Tomography (UGCT), Ghent University, Proeftuinstraat 86, B-9000 Gent, Belgium; 4Laboratory of Pharmaceutical Process Analytical Technology (LPPAT), Department of Pharmaceutical Analysis, Ghent University, Ottergemsesteenweg 460, B-9000 Gent, Belgium; Laurens.leys@ugent.be (L.L.); Gust.nuytten@ugent.be (G.N.); PieterJan.VanBockstal@UGent.be (P.-J.V.B.)

**Keywords:** continuous freeze-drying, µCT, spin-freezing, freezing rate

## Abstract

Spin freeze-drying, as a part of a continuous freeze-drying technology, is associated with a much higher drying rate and a higher level of process control in comparison with batch freeze-drying. However, the impact of the spin freezing rate on the dried product layer characteristics is not well understood at present. This research focuses on the relation between spin-freezing and pore size, pore shape, dried product mass transfer resistance and solid state of the dried product layer. This was thoroughly investigated via high-resolution X-ray micro-computed tomography (µCT), scanning electron microscopy (SEM), thermal imaging and solid state X-ray diffraction (XRD). It was concluded that slow spin-freezing rates resulted in the formation of highly tortuous structures with a high dried-product mass-transfer resistance, while fast spin-freezing rates resulted in lamellar structures with a low tortuosity and low dried-product mass-transfer resistance.

## 1. Introduction

In 2020, the second highest total of novel therapeutics was approved by the FDA’s Center for Drug Evaluation and Research [1]. Almost a quarter of these novel therapeutics are protein-based drugs (e.g., antibodies and antibody–drug conjugates) [1,2]. However, the development of a stable, efficient and safe protein-based drug formulation can be challenging, as many of these proteins lack long-term stability [3,4]. Therefore, cold chain logistics are often required to transfer the biopharmaceuticals from the manufacturing site to the place of administration. The cold chain supply is not only logistically challenging, it is also very costly, and therefore contributes to the soaring costs of healthcare [5]. Therefore, drying techniques are necessary to convert these unstable aqueous formulations into solids of sufficient stability [6,7,8]. Freeze-drying is a frequently used drying technique to enhance the shelf-life of biopharmaceuticals [6,7,8]. However, it is essential to add cryo- and lyo-protectants to the formulation since lyophilisation is associated with freezing and drying stresses, which might impact the stability of biopharmaceuticals. In recent decades, research has been conducted on approaches to minimise degradation during freeze-drying [5,9,10,11,12,13,14,15,16].

Today, freeze-drying is still performed as a batch process, which often lacks cost-effectiveness and thorough quality control during processing [6,8,17,18,19]. Therefore, the development of an innovative drying strategy is imperative. During the last decade, some innovative freeze-drying technologies were proposed [16,18,19,20,21]. Continuous spin freeze-drying is one of these recently developed technologies. This innovative technology uses spin freezing to shorten the total drying time by up to 40 times [18,19,22]. Spin-freezing starts with the rapid rotation of a vial, filled with a liquid formulation, along its longitudinal axis. Subsequently, the temperature of the vial is lowered by a flow of cold, inert gas to induce ice crystal formation. Finally, a thin, cylindrical, frozen product layer is obtained [18,22]. This approach reduces the product layer thickness while the (macroscopic) surface area increases, which results in a significantly higher sublimation rate. Drying is achieved by transferring the spin-frozen product via a load-lock to a drying chamber. A load-lock ensures the transfer of the vial between two chambers with different process conditions without disturbing the process. The product temperature is continuously controlled by an infrared heater in front of every vial in combination with a thermal camera [23,24,25]. Finally, a stable dry product is obtained. Continuous spin freeze-drying does not only result in a shorter drying time, it also offers the possibility of in-line monitoring of critical process parameters and critical quality attributes of every individual vial. This enables the opportunity for real-time feedback control at the individual vial level, which is not possible in a batch freeze-drying process [22]. Another asset of continuous spin freeze-drying in comparison with batch freeze-drying is the ease of up-scaling [18,22,23]. Each vial is subjected to the same process conditions. Therefore, up-scaling can be achieved by simply running the continuous line for a longer time, by making the modules larger or installing multiple parallel lines (i.e., numbering-up). In contrast, batch freeze-drying is associated with complex re-validation procedures since, among others, heat transfer in a small lab-scale freeze-dryer is not comparable to the heat transfer in a production scale freeze-dryer [18,22]. The drying step of this continuous spin freeze-dry technology is currently extensively described in the literature [18,19,22,23,25,26,27]. However, little is known about the impact of the spin-freezing step on the subsequent process steps and critical quality attributes. The spin-freeze technology offers the advantage of controlling each phase of the freezing step. However, the relationship between the size and shape of the pores and cooling rate used during spin-freezing is not yet fully understood. The importance of the freezing step is typically underestimated, while it has been proven for batch freeze-drying that this can have a huge impact on the drying steps [28,29]. During primary drying, an ice-free product layer is formed above the frozen product layer. The diameter of the pores of this dry product layer is dependent on the size and shape of the ice crystals since, after sublimation, a pore is formed with the same shape as the ice crystal [28,29]. Water molecules that have sublimated deeper in the product have to travel through this dry product layer. Consequently, the drying velocity is dependent on the size and shape of the ice crystals. Moreover, the product temperature during primary drying is also related to the pore size since the restriction of water vapour flow, due to small pores, will result in a local increase in water vapour pressure and, therefore, in a higher product temperature [6]. Furthermore, the freezing velocity can also influence the solid state of the formulation. High freezing rates can result in the formation of amorphous excipients, which might not be desired. For example, high cooling rates can result in partly amorphous bulking agents [30]. This is unfavourable since a bulking agent needs to be in the crystalline form to act as a good bulking agent. In addition, a partly amorphous bulking agent will tend to crystallise during storage, which can impact the shelf-life of the product [30]. Furthermore, the solid state of the formulation also influences the reconstitution of the freeze-dried formulation. Reconstitution is determined by a complex interplay of different formulation properties and process settings, such as wettability, protein-to-sugar ratio, cake disintegration rate, cake structure and liquid penetration rate into the cake [31,32,33,34]. The solid state of the dry cake is also an important parameter since partial crystalline cakes tend to have a shorter reconstitution time and a different reconstitution mechanism in comparison to amorphous cakes [31,32,33,34]. Therefore, the solid state of the formulation with respect to the spin-freezing rate will also be evaluated in this research paper.

## 2. Objectives

The spin-freeze technology allows each phase of the freezing step to be controlled. However, the relationship between the size and shape of the pores and cooling rate used during spin-freezing is not yet fully understood. Therefore, the relationship between spin-freezing rate and pore size is thoroughly investigated via high-resolution x-ray micro-computed tomography (µ-CT) and scanning electron microscopy (SEM). Furthermore, as earlier research focused on the determination of the dried product mass transfer resistance ($R_{p}$) of spin freeze-dried product layers via thermal imaging [23], the impact of freezing rate on $R_{p}$ of spin freeze-dried products was not yet evaluated. Here, the information obtained on product structure was used to substantiate the impact of spin-freezing rate on $R_{p}$. Furthermore, the effect of the differences in product structure on the primary drying time was evaluated with thermal imaging. In addition, the influence of spin-freezing on the solid state of the studied formulations was examined via solid-state x-ray diffraction (XRD).

## 3. Methods and Materials

### 3.1. Spin Freezing and Spin Freeze-Drying

Two model formulations were used in this study, a 5% (*w*/*v*) mannitol (Sigma Aldrich, Zwijnaarde, Belgium) solution and a 8% (*w*/*v*) bovine serum albumin (BSA) (Sigma Aldrich, Zwijnaarde, Belgium) solution in deionised water. These formulations were selected because of their differences in solid state after freeze-drying (i.e., crystalline for mannitol and amorphous for BSA). In addition, the solid state of mannitol is dependent on the cooling rate, and is therefore a good model formulation to study the effect of freezing conditions [30].

A 10 mL vial was filled with 3.5 mL of liquid formulation. Next, the vial was placed vertically inside a single-vial spin freeze-dryer (RheaVita, Zwijnaarde, Belgium) (Figure 1). Subsequently, the vial was rapidly rotated around its longitudinal axis at 3500 rpm. Next, the temperature of the vial was lowered by a flow of cold compressed air, cooled using a heat exchanger (part B in Figure 1). The heat exchanger consisted of a stainless steel container filled with liquid nitrogen where a polyurethane tubing (with an internal diameter of 5 mm and total length of 3 m) was submersed.

The temperature of the vial was monitored via thermal imaging (FLIR A655sc, Thermal focus, Ravels, Belgium). In addition, the gas temperature was monitored via a thin gauge type-K thermocouple (Labfacility, Leeds, UK) which was positioned between the vial and the gas outlet. The vial and gas temperature were used as an input for an in-house scripted LABVIEW 2019 (National Instruments, Austin, TX, USA) application to control the freezing step. Cooling of the vial (i.e., before nucleation) was achieved by controlling the vial temperature by changing the mass flow rate of cold compressed air, which was jetted towards the vial. The gas temperature was dependent on the gas flow rate. The gas flow rate was adapted with a mass flow controller (Bronkhorst EL-FLOW Select, Flowcor, Olen, Belgium) via proportional-integral-derivative control using the infrared camera temperature measurements.

After nucleation, the spin-freezing rate was controlled by keeping the heat flow rate ($\dot{Q}$) constant. This can be achieved, as represented in Equation (Equation 1), by adapting the gas flow rate in the function of the vial temperature, which was recorded via thermal imaging.
(1)$$\dot{Q}=A_{vial}k\left(T_{vial}-T_{gas}\right)$$
where $A_{vial}$ is the vial surface area (m${}^{2}$), *k* the heat transfer coefficient during freezing (W/m${}^{2}$ K), $T_{vial}$ the vial temperature (K) and $T_{gas}$ the gas temperature (K). The heat transfer coefficient was calibrated beforehand via linear regression for multiple gas flow rates. During the freezing phase (i.e., after nucleation), the vial and gas temperature were continuously monitored. These parameters were used as an input in the LABVIEW script. Consequently, the gas flow rate was continuously adapted to change the heat transfer coefficient (*k*) and to maintain the desired $\dot{Q}$. The cooling phase prior to nucleation is linked to the freezing phase; therefore, low cooling rates will be accompanied by slow freezing rates and vice versa.

Two different freezing rates were used during spin-freezing, a slow freezing rate (4 °C/min during cooling of the liquid (i.e., before nucleation) and $\dot{Q}$ = 1.4 W during freezing) and a fast freezing rate (50 °C/min during cooling of the liquid (i.e., before nucleation) and $\dot{Q}$ = 17.0 W during freezing). In the single-vial drying chamber, a vacuum was applied as soon as the vial obtained a temperature of −50 °C. Each vial was dried at 10 Pa without the use of an infrared heater, while rotating at 5 rpm (i.e., drying was based on the radiation from the surroundings). This is a very conservative and non-optimised drying cycle to avoid (micro-)collapse during drying, which might influence the size and shape of the pores. After 5 h, the spin freeze-dried vial was stoppered under vacuum. Subsequently, the single-vial drying chamber was aerated with inert nitrogen gas to end the drying procedure (process parameters represented in Table 1).

### 3.2. Determination of Collapse Temperature

Freeze-drying microscopy was used to determine the collapse temperature of the formulations. An optical microscope (BX51, Olympus, Hamburg, Germany) was equipped with an FDCS 196 freeze-drying module (Linkam, Surrey, UK). A quartz sample holder was placed on top of a temperature-controlled block of the freeze-drying module. Next, 2 µL of liquid formulation was carefully placed on the quartz sample holder. Subsequently, the liquid sample was covered with a glass coverslip. A thin metal U-shaped spacer was placed between the quartz sample holder and glass coverslip to provide a way for vapour escape. Finally, the freeze-drying stage was hermetically closed with an aluminium lid. Freezing was started by lowering the temperature of the freezing module to −50 °C at a rate of 1 °C/min. This temperature was maintained for 15 min to avoid a temperature gradient in the sample. Next, the pressure of the freeze-drying module was lowered to 1 Pa with a vacuum pump (E2M1.5, Edwards, Nazareth, Belgium). Subsequently, the temperature of the product was increased every minute by 1 °C. Meanwhile, the product structure was monitored via the optical microscope. The formation of ruptures in the dried product was assumed to be the onset of collapse, and the product temperature at the moment of rupture formation was defined as the collapse temperature.

### 3.3. High Resolution X-ray Micro-Computed Tomography

High-resolution X-ray micro-computed tomography (µCT) was used to evaluate the pore size and product structure on a micro-scale. Sample preparation was done in a glovebox under a dry nitrogen atmosphere to avoid moisture uptake. A sample of ±1 cm${}^{3}$ was taken from each freeze-dried formulation and stored in a sealed plastic container to avoid contact with the atmosphere during scanning. Next, the closed container was vertically positioned in a custom-built high-resolution X-ray micro-computed tomography scanner (HECTOR, UGCT, Ghent, Belgium) [35]. The isotropic reconstructed voxel size and other relevant scanning information are represented in Table 2. The tube was operated at a power of 10 W and voltage of 80 kV for the mannitol sample with fast spin-freezing rate and 90 kV for the other samples. For a full 360° rotation, 2401 projections were taken at an exposure time of 1 s per image. Tomographic reconstructions were made using Octopus Reconstruction software [36]. First, all reconstructions were qualitatively evaluated. Next, the most interesting and relevant regions were quantitatively analysed with Octopus Analysis [37]. First, a specific threshold (i.e., different formulations are characterised by different thresholds) was applied to separate air from solid material (i.e., BSA or mannitol) in each selected 3D region. The thresholds were selected based on visual evaluation of the separation between air and solid material. Three thresholds were included in the analysis to evaluate the effect on the pore size calculations. Next, labelling and watershed separation was used to identify the individual pores. Different descriptors were used to analyse the size of the pores. The maximum opening diameter (MO) was defined as the diameter of the largest sphere that fits into the pore, while the equivalent diameter (ED) is defined as the diameter of a sphere containing the same amount of voxels as the pore [37]. In addition, the hydrodynamic diameter (HD) is calculated as:(2)$$HD=\frac{4V_{pore}}{A_{pore}}$$
where $V_{pore}$ is the volume of the analysed pore (m${}^{3}$) and $A_{pore}$ the surface area of the analysed pore (m${}^{2}$).

Next to the pore size, the tortuosity along the Z-axis of the sample was calculated via the Octopus analysis software. First, the connections calculated in the watershed based separation were used as an input to determine a skeleton. The centrepoint of each watershed was connected by a straight line. In this way, a basic skeleton was obtained that can be used to calculate the tortuosity, which was defined as the ratio of the geodesic distance to the euclidean distance with respect to the height of the product [36,37]. To visualize the samples in virtual 3D volume, the stack of 2D images was rendered using VGSTUDIO MAX (Volume Graphics, Heidelberg, Germany).

### 3.4. Scanning Electron Microscopy

Scanning electron microscopy (SEM) was used to evaluate the surface and longitudinal cross section of the freeze-dried cakes (Quanta 200F, Thermo Fisher Scientific, Waltham, MA, USA). Sampling was achieved by tapping the spin freeze-dried vial gently on the bench to cause breakage of the cake in order to avoid artefacts caused by scalpel cutting. Next, ±1 cm${}^{2}$ of the freeze dried cake was attached to a copper sample holder with carbon tape. Next, the electron conductivity of the samples was improved by sputtering each sample with a 40 nm gold coating using a sputter coater (Quorum Technologies, Laughton, UK). Finally, the product structure was evaluated in the scanning microscope under high vacuum conditions (i.e., 10${}^{-4}$ Pa).

### 3.5. Determination of Dried Product Mass Transfer Resistance

As primary drying proceeds, a dry porous product layer is obtained. The pore size and tortuosity of this product layer has a huge impact on the total drying time. This is caused by the restriction of water vapour flow through the dried layer. This water vapour flow restriction is associated with a local increase in water vapour pressure at the sublimation interface ($P_{i}$). Consequently, the temperature at the sublimation interface ($T_{i}$) will also increase as $T_{i}$ is in equilibrium with $P_{i}$ according the Clausius-Clapeyron relation. However, $T_{i}$ should remain below the collapse temperature during drying to avoid collapse and, thus, loss of product structure. Therefore, the mass transfer resistance caused by the dried product layer ($R_{p}$) is a crucial parameter for the development of a freeze-drying cycle. $R_{p}$ (m/s) was determined as:(3)$$R_{p}=\frac{A_{p}\left(P_{i}-P_{c}\right)}{{\dot{m}}_{sub}}$$
where $A_{p}$ is the frozen product area (m${}^{2}$), $P_{i}$ the vapour pressure of ice at the sublimation front (Pa), $P_{c}$ the partial water vapour pressure in the drying chamber (Pa) and ${\dot{m}}_{sub}$ the sublimation rate (kg/s) [22,25]. During primary drying, the drying chamber is mainly filled with water vapour. Therefore, it was assumed that the partial water vapour pressure is equal to the overall pressure in the drying chamber. In addition, it was assumed that the process is in steady-state conditions during primary drying. Therefore, all the energy (i.e., total power ($P_{tot}$)) was assumed to be used for sublimation. Therefore, ${\dot{m}}_{sub}$ was calculated according to Equation (Equation 4).
(4)$${\dot{m}}_{sub}=\frac{P_{tot}M}{\Delta H_{sub}}$$
where $\Delta H_{sub}$ is the latent heat of sublimation (J/mol) and *M* the molecular weight of water (Kg/mol). Thermal imaging was used to determine $T_{i}$ by monitoring the vial temperature during drying. The vial temperature was converted to $T_{i}$ by using the Fourier law to correct for the temperature gradient over the glass vial and frozen product layer [23]. The temperature at sublimation front was hence converted into $P_{i}$ according to the Clausius–Clapeyron relation:(5)$$P_{i}=3.6\times 10^{12}\;e^{-\frac{6145}{T_{i}}}$$

The frozen product area ($A_{p}$) was calculated according to:(6)$$A_{p}=2\pi \left(\sqrt{r_{v}^{2}-\frac{V_{fill}}{\pi h}}+L_{dr}\right)h$$
where $r_{v}$ is the radius of the vial (m), $V_{fill}$ is the filling volume (m${}^{3}$), *h* the height of the frozen product layer (m) and $L_{dr}$ the dried layer thickness (m). The dried layer thickness ($L_{dr}$) was calculated using Equation (Equation 7).
(7)$$L_{ice}=L_{tot}-L_{dr}=\frac{\Phi V_{fill}\rho_{sol}}{A_{p}\rho_{ice}}-\frac{{\dot{m}}_{sub}\Delta t}{\Phi A_{p}\rho_{ice}}$$
where $L_{ice}$ is the ice layer thickness (m), $L_{tot}$ the total layer thickness (m), Φ the porosity (-), $\rho_{sol}$ the density of water (kg/m${}^{3}$) and $\rho_{ice}$ the density of ice (kg/m${}^{3}$).

### 3.6. Evaluation of the Primary Drying Time

A thermal camera (FLIR A655sc IR camera, Thermal Focus, Ravels, Belgium) was used to monitor the vial temperature during primary drying through a germanium window. Van Bockstal et al. [23] described that a steep increase in vial temperature is a good marker for the end of primary drying. The primary drying time was determined for each formulation and each freezing-rate.

### 3.7. X-ray Diffraction

The solid state of the freeze-dried formulations was determined via X-ray diffraction (XRD). XRD patterns were recorded in the angular range 5 to 80° (2θ) using a step size of 0.040° and counting time of 1 s/step. The XRD scanner was equipped with a copper source (Siemens, Karlsruhe, Germany) with a voltage of 40 kV. Approximately 0.5 g of each freeze-dried formulation was gently crushed and positioned in the sample holder.

## 4. Results and Discussion

### 4.1. Determination of Collapse Temperature

The collapse temperature of both formulations was evaluated via freeze-drying microscopy. Rupture formation in the freshly formed dried layer was interpreted as the onset of collapse. The first signs of loss in product structure was observed at −9 °C for the BSA formulation. A porous and uninterrupted structure was again obtained as soon as the product temperature was lowered to −10 °C. For the mannitol formulation, the formation of ruptures in the dried product layer was observed at −2 °C (i.e., fusion). Therefore, it was concluded that the collapse temperature of the BSA formulation and mannitol formulation was −9 and −2 °C, respectively. These are rather high collapse temperatures, which minimise the risk of collapse during primary drying.

### 4.2. High-Resolution X-ray Computed Tomography

The pore size and shape were thoroughly evaluated via high-resolution X-ray computed tomography, as described in Section 3.3. First, the results were qualitatively evaluated via reconstructed 3D-images, as represented in panel A and B of Figure 2 and Figure 3. For both mannitol and BSA, the high spin-freezing rate resulted in low tortuous product structures, characterised by elongated, lamellar, straight and very regular pores with a small diameter (referred to as chimney-like pores). On the other hand, the low spin-freezing rate was associated with the formation of a more tortuous structure with rather irregular pore shapes and apparently larger pores in comparison with the pores obtained via high spin-freezing rates. However, local variations in pore structures were observed for both cooling rates and both formulations. This is consistent with the earlier work of the authors, where local variations in mass transfer resistance were linked to possible local variations in pore structure [25].

Quantitative analysis was achieved by comparing different pore size descriptors, as depicted in panels C-F of Figure 2 and Figure 3 (for mannitol and BSA, respectively). The equivalent diameter (ED) resulted in the largest pore sizes as it was dependent on the volume of the pores and the assumption that all pores are perfect spheres, and hence did not account for the elongation of the pores. The distribution based on the maximum opening diameter (MO) is strongly affected by discretization effects, and is therefore less reliable. The hydrodynamic diameter (HD) takes the pore-volume and pore-area into account, as described in Equation (Equation 2). The distribution based on HD was in agreement with the pore size estimated from the SEM-images, and is thus less prone to such discretization effects. It was, therefore, selected as the most relevant diameter to describe the pore size. The labelling and segmentation of each 3D section, to separate air from material (i.e., mannitol or BSA), was performed with three different gray-value thresholds (and thus attenuation coefficients) to evaluate the impact on the distribution. An increase in hydrodynamic diameter was observed when a lower spin-freezing rate was used (for both bsa and mannitol), independent of the attenuation coefficient. A reversed relation between pore size and cooling rate was expected, as this is already widely described in the literature [6,9,28,29,34,38]. However, as no clear relation between degree of supercooling and pore size was found during this study, this should be further examined in future research.

Next to pore size, the tortuosity along the Z-axis (i.e., product layer thickness) of the spin freeze-dried cakes was determined via µCT. As depicted in Figure 4, an inverse relationship was found between the spin-freezing rate and tortuosity. The tortuosity values were statistically different, as verified with a one-way ANOVA test (*p*-value = 0.00010, at 95% significance level).

A high spin-freezing rate reduced the tortuosity of the spin freeze-dried cake compared to the tortuosity of spin freeze-dried cakes produced via low cooling rates. This is consistent with the elongated, lamellar pore structures, which were observed during the qualitative analysis of the 3D-images of the spin freeze-dried cakes produced via high spin-freezing rates. These findings were valid for both formulations. However, a lower tortuosity level was found for the BSA formulation compared to the mannitol formulation. This was possibly caused by the formation of cracks in the freeze-dried cake. These cracks were visually observed in the cakes of the BSA formulation and were also observed during the authors’ earlier work [25]. Crack formation is a result of the release in tension, which is built up during drying [39]. These cracks disrupt the pore network, possibly reducing the tortuosity.

### 4.3. Scanning Electron Microscopy

Next to µCT, SEM was used to evaluate the effect of the spin-freezing conditions on pore size and tortuosity. As represented in Figure 5 and Figure 6, SEM-analysis revealed lamellar pore structures at the longitudinal cross-sections of the BSA and mannitol freeze-dried cakes produced with fast spin-freezing conditions. These structures with low tortuosity (i.e., the lamellar structures) were also observed via µCT and are typically related to high freezing rates [28,34]. Furthermore, elongated, regular but small pores were observed in the SEM-images of the surface of the freeze-dried products produced via fast spin-freezing conditions. The lamellar structures are probably a result of directional solidification, where the fast freezing rates inhibit the formation of side branches during ice crystal formation, which results in the formation of ice spherulites [28,34]. However, SEM-images of the radial cross-sections of freeze-dried product with slow freezing conditions revealed the existence of irregular, high-tortuous pore structures. It can be assumed that, if the freezing rate is slow enough, ice crystals have the time to form side branches. As a result, ice can form stable hexagonal dendrites [28,34] and a highly tortuous network is formed.

### 4.4. Determination of Dried Product Mass Transfer Resistance

The dried product mass transfer resistance was determined as described in Section 3.5. As depicted in Figure 7, the cooling rate used during spin-freezing had a major impact on $R_{p}$. However, the $R_{p}$ values of the mannitol formulation are very similar for each cooling rate during the first part of primary drying (Figure 7A). This is in correspondence with the similar pore sizes obtained after fast or slow spin-freezing. In the first part of primary drying, the dried-product layer thickness is rather limited, and is hence characterised by a low tortuosity. As primary drying proceeds, the dried layer thickness increases, causing an increase in $R_{p}$ due to an increased tortuosity. SEM-analysis and µCT-analysis revealed that high spin-freezing rates resulted in lamellar pores with low tortuosity. These chimney-like structures are beneficial for water vapour flow through the product (and causes lower tortuosity values). Hence, lower $R_{p}$ values are maintained throughout the formation of the dry layer despite the smaller size of the pores. In contrast, slow freezing rates resulted in a more tortuous product structures and, therefore, in higher $R_{p}$ values.

The BSA formulations were generally associated with lower $R_{p}$ values (Figure 7B) in comparison with the mannitol formulations. This can be linked to the slightly larger pores in the cakes of the BSA formulations. Larger pores are associated with a more efficient water vapour transport and, thus, a lower $R_{p}$ value. This can also be explained by the cracks observed in the cake of BSA-based formulations. This was also observed during the earlier work of the authors where the formation of cracks in freeze-dried cakes was associated with a local increase in sublimation velocity [25]. The formation of these cracks is a result of the release of tension that is built up during drying [39]. The formation of cracks is mainly dependent on the solid state of the material and the mechanical strength of the cake. Mannitol is a typical bulking agent, which is less sensitive to the formation of cracks. Therefore, the lack of cracks in the mannitol-based cake also supports this hypothesis.

As extensively discussed in the literature, $R_{p}$ is not only determined by freezing rate, but also by the total solute concentration. Higher solute concentrations are related to smaller pores, which results in a higher $R_{p}$ [23,25,40,41]. However, the effect of a higher solid concentration on the mass transfer resistance was not evaluated in this study, as this will be part of future research.

It must be noted that $R_{p_{0}}$, which is determined as the value of the $R_{p}$ curve at zero thickness, is significantly different from 0. This is also reported in the literature, and is possibly due to the formation of a low permeable skin (consisting of densified freeze concentrate) at the top of the cake at the ice surface [19,38,42,43]. At present, the formulation and processing factors that contribute to skin formation are unclear and need to be investigated further [19,38,42,43].

### 4.5. Evaluation of Primary Drying Times

Large differences in dried-product mass -ransfer resistance were observed when the freezing rate was altered during spin freezing. However, spin freeze-drying is characterised by rather short primary drying times and thin dried-product layers, especially when compared with batch freeze-drying. Therefore, the impact of the cooling rate on primary drying time was evaluated as described in Section 3.6. As represented in Figure 8, a total increase in primary drying time of 7 and 5% for the BSA and mannitol formulation, respectively, was observed via thermal imaging when a slower spin-freezing rate was used. The primary drying times are statistically different from each other, as verified with a one-way ANOVA test (*p*-value = 0.0010, at 95% significance level). Despite these statistical differences, the maximum difference in primary drying time was only 8 and 6 min for the BSA and mannitol formulation, respectively. Therefore, it can be concluded that the freezing rate only had a limited impact on the primary drying time despite the differences in $R_{p}$. However, the cooling of vials via a cryogenic gas allows for a much wider range of freezing rates in comparison with batch freeze-drying. Therefore, the impact of such cooling rates on protein stability is questioned [44]. In addition, the drying step in spin freeze-drying also significantly differs from batch freeze-drying [19,22,26]. Drying is no longer based on conduction via temperature-controlled shelves but is achieved via infrared (IR) radiation. However, the impact of IR radiation on the long-term stability of proteins is unknown. Therefore, an additional study is required to evaluate the (long-term) stability of proteins after spin-freeze drying.

### 4.6. X-ray Diffraction

The solid state of the material was evaluated via XRD, as described in Section 3.7. As depicted in Figure 9, no clear peaks could be detected in the X-ray diffraction pattern of BSA, which indicates that the BSA formulation is amorphous after spin freeze-drying, independent of the spin-freezing condition. In contrast, XRD of the mannitol formulation resulted in a clear X-ray diffraction pattern, which indicates that the mannitol formulation is crystalline after spin freeze-drying. As the X-ray diffraction patterns of the slow and fast spin-frozen mannitol formulation overlap, the chosen spin-freeze conditions did not alter the solid state of the mannitol formulations. The X-ray diffraction patterns were compared with the characteristic peaks of the different mannitol polymorphs (i.e., anhydrous α, anhydrous β, anhydrous δ and hemihydrate mannitol) found in the literature [30]. The characteristic peaks of δ-mannitol (i.e., at 9.94 and 24.64°) and β-mannitol (i.e., at 14.56 and 16.76°) were identified in the X-ray diffraction patterns. No characteristic peaks of mannitol hemihydrate (at 9.6, 16.5, 18 and 25.7°) or anhydrous α-mannitol (at 13.7 and 17.3°) were detected. The absence of mannitol hemihydrate is beneficial since it is prone to transforming into anhydrous crystalline forms during storage. This transformation from hemihydrate to anhydrous mannitol is associated with the release of lattice water, which might impact the protein stability and, therefore, could shorten the product shelf life [30].

## 5. Conclusions

The impact of the spin-freezing rate on pore size, pore shape and tortuosity of the dried-product layer obtained after spin freeze-drying was evaluated via quantitative X-ray µCT and SEM. Two model formulations were used, one based on mannitol (i.e., a crystalline formulation) and one based on BSA (i.e., an amorphous formulation). It was concluded that a higher spin-freezing rate resulted in small, elongated, lamellar pores characterised by a low tortuosity. In contrast, a low spin-freezing rate resulted in larger, more spherical pores with a higher tortuosity. These findings were in agreement with the freezing mechanisms described for batch freeze-drying. It was stated that fast freezing rates resulted in directional solidification and the formation of lamellar ice structures [14,28,34,45].

Wheras earlier research focused on the determination of $R_{p}$ in spin freeze-dried product layers via thermal imaging [23], the impact of freezing rate on $R_{p}$ in spin freeze-dried products was not evaluated. Here, $R_{p}$ determination based on thermal imaging revealed a reduction in $R_{p}$ at high spin-freezing rates (due to the lower tortuosity). However, only small differences in primary drying time were observed when the spin-freezing rate was altered. In addition, the solid state of both formulations in the function of the spin-freezing rate showed that the BSA formulation was amorphous after spin freeze-drying independent of the cooling rate. In contrast, the mannitol formulation was crystalline after freeze-drying, showing characteristic peaks of δ-mannitol and β-mannitol polymorphs independent of the cooling rate.

## Figures and Tables

**Figure 1 pharmaceutics-13-02126-f001:**
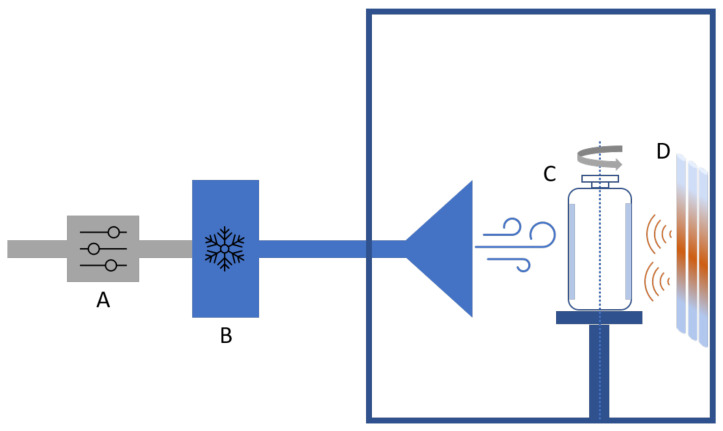
Schematic representation of spin freeze-drying. (**A**) Mass flow controller (**B**) Heat exchanger (**C**) Rotating vial (**D**) Infrared heater.

**Figure 2 pharmaceutics-13-02126-f002:**
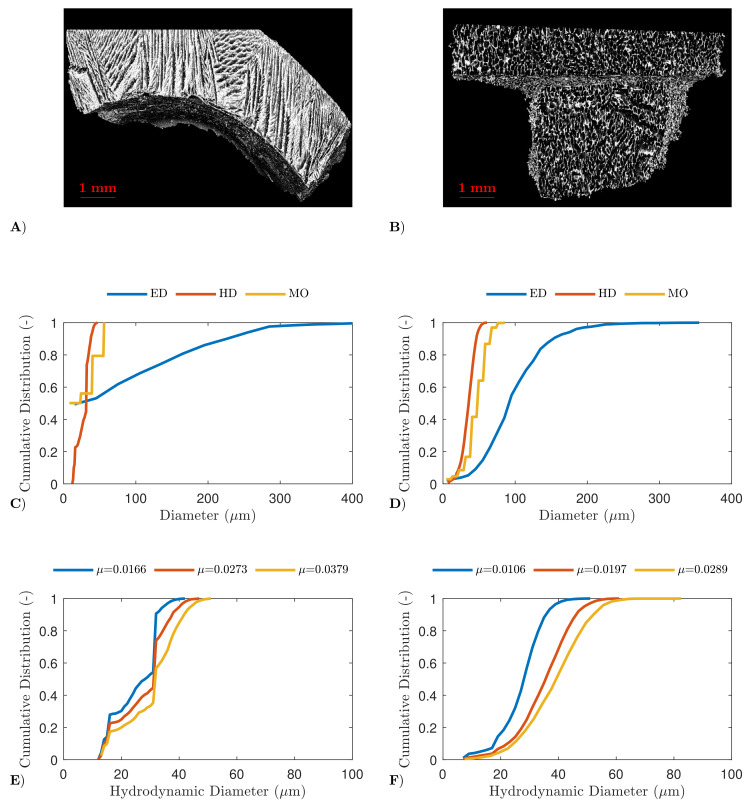
µCT results of the mannitol formulation. Reconstructed 3D image of the dried product layer obtained via a high (panel **A**) and slow spin-freezing rate (panel **B**). The cumulative distribution of the equivalent diameter (ED), hydrodynamic diameter (HD) and maximum opening (MO) of the dried product layer obtained via a high (panel **C**) and slow spin-freezing rate (panel **D**). The cumulative distribution of the hydrodynamic diameter of the dried product layer obtained via a high (panel **E**) and slow spin-freezing rate in function of used threshold (panel **F**).

**Figure 3 pharmaceutics-13-02126-f003:**
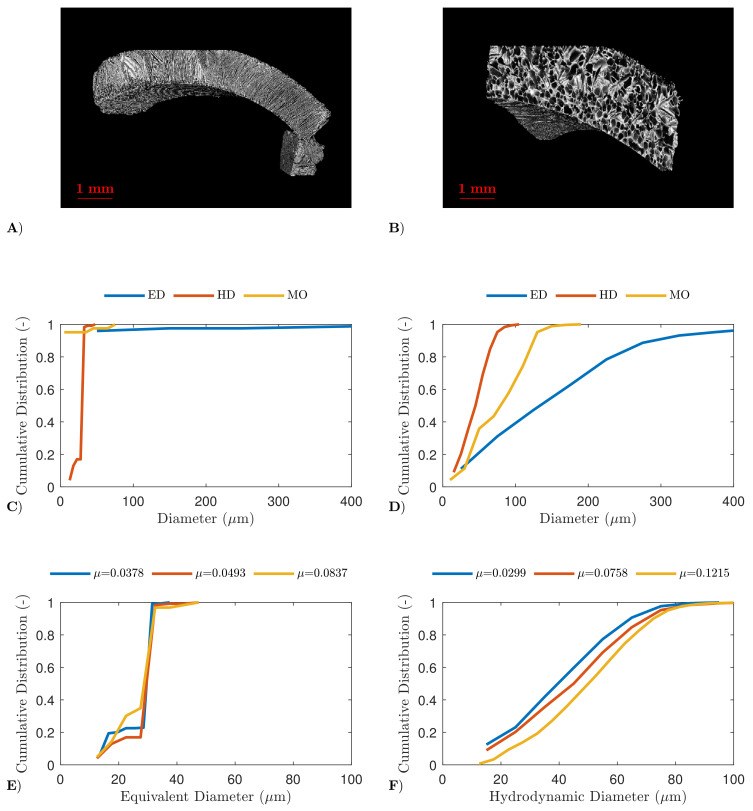
µCT results of the BSA formulation. Reconstructed 3D-image of the dried product layer obtained via a high (panel **A**) and slow spin-freezing rate (panel **B**). The cumulative distribution of the equivalent diameter (ED), hydrodynamic diameter (HD) and maximum opening (MO) of the dried product layer obtained via a high (panel **C**) and slow spin-freezing rate (panel **D**). The cumulative distribution of the hydrodynamic diameter of the dried product layer obtained via a high (panel **E**) and slow spin-freezing rate in function of used threshold (panel **F**).

**Figure 4 pharmaceutics-13-02126-f004:**
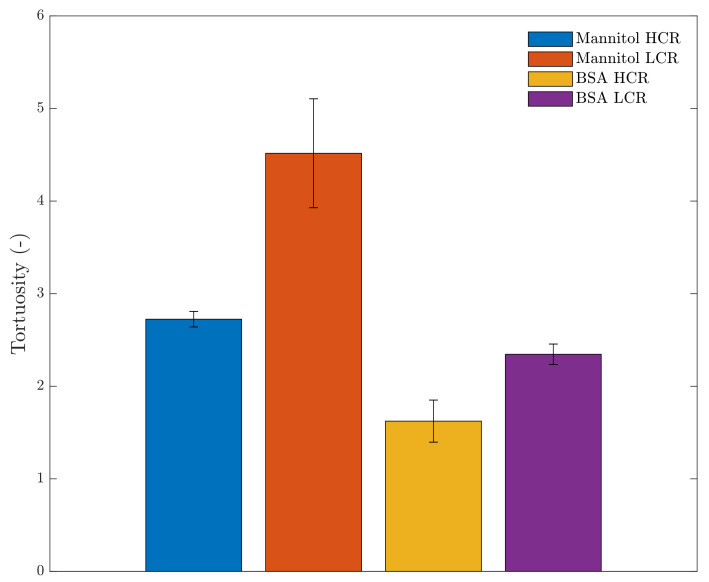
Tortuosity of the dried product layer along the Z-axis per sample determined via µCT: mannitol produced via a high spin-freezing rate (blue); mannitol produced via a low spin freezing rate (orange); BSA produced via a high spin-freezing rate(yellow); BSA produced via a low spin-freezing rate (purple).

**Figure 5 pharmaceutics-13-02126-f005:**
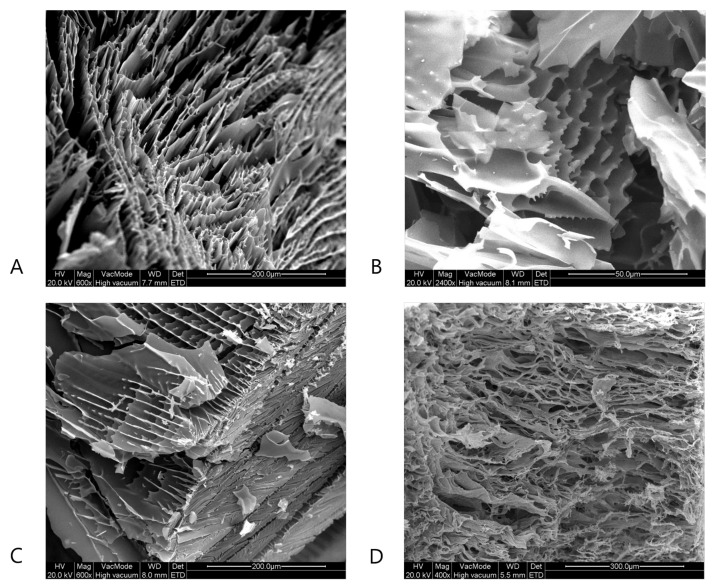
SEM-images of the dried product layer of the mannitol formulation for a high spin-freezing rate (panel **A** (cross-section) and **C** (surface)) and a low spin-freezing rate (panel **B** (cross-section) and **D** (surface)).

**Figure 6 pharmaceutics-13-02126-f006:**
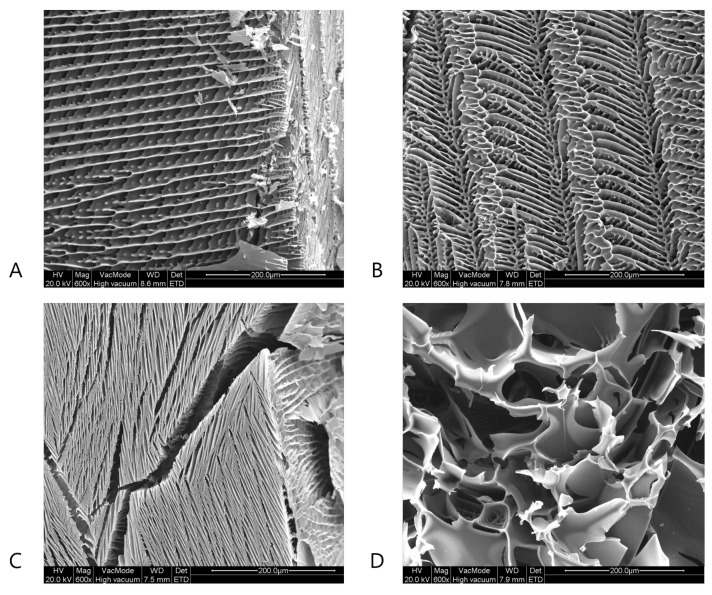
SEM-images of the dried product layer of the BSA formulation for a high spin-freezing rate (panel **A** (cross-section) and **C** (surface)) and a low spin-freezing rate (panel **B** (cross-section) and **D** (surface)).

**Figure 7 pharmaceutics-13-02126-f007:**
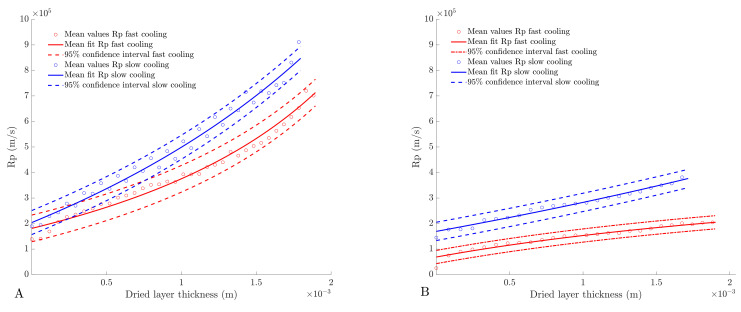
(**A**) Dried-product mass-transfer resistance ($R_{p}$) determined via thermal imaging for fast (red) and slow spin-freezing (blue) of the mannitol formulation (**B**) Dried product mass transfer resistance ($R_{p}$) for fast (red) and slow spin-freezing (blue) of the BSA formulation.

**Figure 8 pharmaceutics-13-02126-f008:**
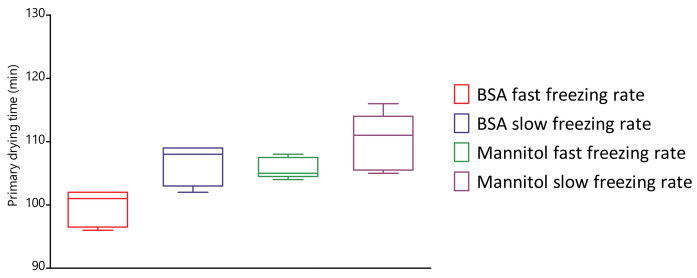
Boxplot of the primary drying time determined via thermal imaging (whiskers indicate the 10–90 percentile). A total increase in primary drying time of 7 and 5% for the BSA and mannitol formulation, respectively, was observed via thermal imaging when a slower spin-freezing rate was used (statistically different from each other, *p*-value = 0.0010, at α = 95%).

**Figure 9 pharmaceutics-13-02126-f009:**
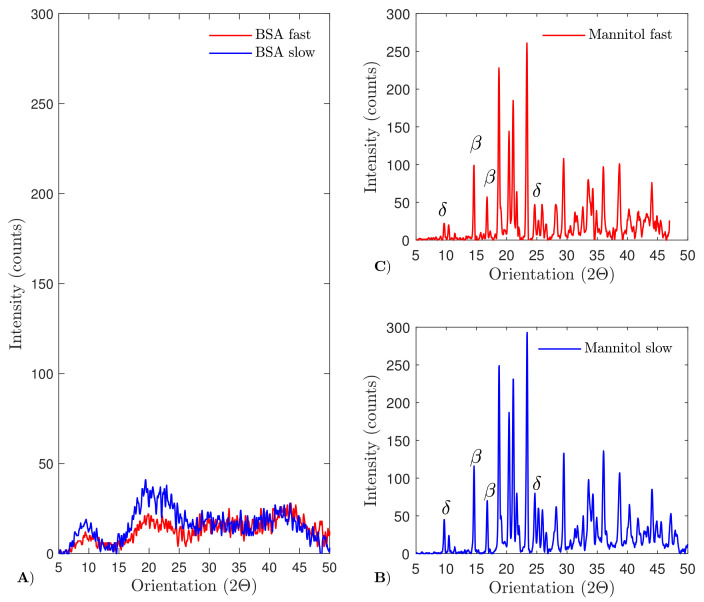
X-ray diffraction patterns of BSA (panel **A**) and mannitol (panels **B** and **C**). No clear diffraction pattern was obtained after measuring the BSA-based samples, indicating that the BSA formulation was amorphous after spin freeze-drying. The X-ray diffraction patterns of the slow (panel **B**) and fast (panel **C**) spin-frozen mannitol formulation are identical and show characteristic peaks of δ-mannitol and β-mannitol polymorphs.

**Table 1 pharmaceutics-13-02126-t001:** Process parameters during spin-freeze drying.

Sample	Cooling Rate (°C/min)	Freezing Rate (W)	Chamber Pressure (Pa)	Total Drying Time (h)
Mannitol fast	50	17	10	5
Mannitol slow	4	1.4		
BSA fast	50	17		
BSA slow	4	1.4		

**Table 2 pharmaceutics-13-02126-t002:** Settings per µCT scan.

Sample	Voxel Size (µm)	Projections (-)	Analysed Volume (mm${}^{3}$)
Mannitol fast	8	2401	4.6
Mannitol slow	4.5	2601	4.55
BSA fast	8	2401	4.6
BSA slow	8	2401	4.14

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
