# Peer review of "Spin Freezing and Its Impact on Pore Size, Tortuosity and Solid State"

_pharmaceutics, 2021, doi:10.3390/pharmaceutics13122126_

Round 1

Reviewer 1 Report

The paper is written logically and based on a robust basis of experiments. the topic is of interest. Spin freezing is an emerging technology that deserves investigation. I have included in the attached pdf text comments and suggestions for the authors to fix.

Reviewer 2 Report

The authors investigated and evaluated the effect of spin freezing on the physicochemical properties of products. In their study, a definite amount of mannitol and BSA were dissolved together. If there are any data about the other concentrations, the author should show and discuss them. And, the stability of amorphized BSA should be described.

Reviewer 3 Report

The paper “Spin freezing and its impact on pore size, tortuosity and solid state” by Lammens at al. deals with the effect of different spin freezing rates on the morphology of the resulting freeze dried product, as well as on process performance (in terms of primary drying time). A number of experimental techniques, including micro-CT, SEM, XRD and infrared thermography are used for this purpose.

Overall, the paper is clear and nicely written. The experiments have been accurately performed, and the results are of interest for the freeze-drying community. I would therefore recommend publication of this work, after the following points have been taken into account:

Abstract:

“It was concluded that slow spin freezing rates resulted in the formation of small pores […]” actually, in the remainder of the work, slow spin freezing rates are associated with larger pores, but showing high tortuosity values that are detrimental for water vapor removal by sublimation. Please, change ‘small pores’ with ‘highly tortuous structures’ or similar, to be consistent with the rest of the work.

Introduction:

“A load-lock ensures the transfer of the vial between two chambers which different process conditions without disturbing the process” ‘which’ should probably be ‘with’

Methods:

“Then, the temperature of the vial was lowered by a flow of cold compressed air.” what was the temperature of cold compressed air?

“Two different freezing rates were used during spin freezing, one slow freezing rate (4 °C/min) and one fast freezing rate (50 °C/min).” This sentence is unclear. The authors previously declare they controlled the heat flow rate Q during freezing. This makes sense, because the temperature of a vial undergoing freezing would tend to remain close to the equilibrium freezing value until solidification is complete, and defining a freezing rate in °C/min would therefore be extremely hard while freezing is occurring. But now the authors talk about freezing rate in °C/min. Please clarify.

What was the cooling rate employed during freezing in the freeze drying microscope experiments (section 3.3)?

Results and discussion:

section 4.1: for mannitol, which is crystalline, it would be better to talk about fusion, rather than collapse. Collapse can be more correctly attributed only to amorphous products.

How was tortuosity estimated from the micro-CT images? Please add some details about the procedure employed.

Reviewer 4 Report

The article entitled “Spin freezing and its impact on pore size, tortuosity and solid state” focuses on the relation between spin freezing and pore size, pore shape, dried product mass transfer resistance and solid state of the dried product layer. The basic concept is interesting, however significant changes are required before further decision. My comments regarding to the content are the followings:

Please modify the form of your manuscript according to the requirements of Pharmaceutics template journal!

The role and types of cryoprotectants is not mentioned in the introduction part, however their application is crucial is case of protein lyophilization.

It is not clear what is the novelty of work! Please clarify it!

Please rearrange the methods section, the applied BSA-mannitol solution and its preparation belongs to rather the freeze-drying method description, but for sure not to “materials” section.

Please provide a figure about the spin freeze-dryer, how it operates, why is it different from conventional batch freeze-dryer.

Operation parameters should be added to a separate table to make it more clear for the reader what was the difference between fast and slow drying cycle.

Statistical analysis should be carried out the investigate significance tortuosity of the dried products (Fig. 3)

Figure 8. hard to interpret for the reader as the diffractograms overlap with each other and also maybe different colours for fast and slow freeze-dried materials can be used to make more clear the differences in characteristic peaks in the diffractogram. The diffractogram of plain materials would be also important to present to support which drying process resulted in polymorphic transition.

The conclusion does not contain any literature reference, which support the significance of the authors results. What is the main finding?

Round 2

Reviewer 4 Report

The Authors have addressed all of my concerns with their response to reviewers questions. The revised manuscript was significantly improved, in present form it can be accepted for publication.